# A Real-Time Adaptive Station Beamforming Strategy for Next Generation Phased Array Radio Telescopes

**DOI:** 10.3390/s24144723

**Published:** 2024-07-20

**Authors:** Guoliang Peng, Lihui Jiang, Xiaohui Tao, Yan Zhang, Rui Cao

**Affiliations:** Key Laboratory of Aperture Array and Space Application, East China Research Institute of Electronic Engineering, Hefei 230088, China; pengguol@163.com (G.P.); emanonxmu@sina.com (L.J.); leotau@outlook.com (X.T.); hfutzhangyan@163.com (Y.Z.)

**Keywords:** phased array, radio telescope, adaptive beamforming, spatial filtering, Least Mean Square (LMS), real-time parallel processing

## Abstract

The next generation phased array radio telescopes, such as the Square Kilometre Array (SKA) low frequency aperture array, suffer from RF interference (RFI) because of the large field of view of antenna element. The classical station beamformer used in SKA-low is resource efficient but cannot deal with the unknown sidelobe RFI. A real-time adaptive beamforming strategy is proposed for SKA-low station, which trades the capability of adaptive RFI nulling at an acceptably cost, it doesn’t require hardware redesign but only modifies the firmware accordingly. The proposed strategy uses a Parallel Least Mean Square (PLMS) algorithm, which has a computational complexity of 4N+2 and can be performed in parallel. Beam pattern and output SINR simulation results show deeply nulling performance to sidelobe RFI, as well as good mainlobe response similar to the classical beamformer. The convergence performance depends on the signal-and-interference environments and step size, wherein too large a step size leads to a non-optimal output SINR and too small a step size leads to slow convergence speed. FPGA implementation demonstrations are implemented and tested on a NI FPGA module, and test results demonstrate good real-time performance and low slice resource consumption.

## 1. Introduction

The next generation phased array radio telescope, such as the Square Kilometre Array (SKA) [1], the Low Frequency Array (LOFAR) [2], the Long Wavelength Array (LWA) [3], the Mileura Wide-Field Array (MWA) [4] and the 21 CentiMeter Array (21CMA) [5], consists of a large number of antenna element forming a very large aperture. The SKA is the world’s largest phased array radio telescope with an aperture of up to one square kilometer, which covers two different frequency ranges and are named to reflect this: SKA-Mid, an array of 197 traditional dish antennas (350 MHz to 15.35 GHz), and SKA-Low, an array of 131,072 smaller tree-like antennas (50 MHz to 350 MHz) [6]. SKA-Low’s antennas are divided into 512 stations, with 256 antennas per station. By applying individual per-channel phase coefficient to each antenna, the SKA-low can digitally “points” in one or multiple directions in the sky flexibly, however wide fields of view for tree-like antennas make them highly susceptible to RF interference (RFI) from horizon to horizon [7].

Although SKA-low aperture array is built in radio-quiet zone, where several levels of protection are in place to limit the RF Interference (RFI), but not all terrestrial transmissions can be prevented, such as interference from satellite transmitters or accidental electromagnetic radiation. A variety of RFI events, such as 146 MHz variable RFI, broad-band DTV RFI and 150.17 MHz short RFI burst have been observed at the site of SKA-low [8]. These RFI may corrupt the astronomical signal of interest. There are three main options of RFI mitigation methods in radio astronomy observations: temporal domain rejection, frequency rejection and spatial filtering. Of these, spatial filtering is an RFI mitigation method unique to phased array radio telescope, which use the difference in the direction-of-arrival (DOA) of the astronomical signal-of-interest (SOI) and the RFI [9].

SKA-low adopts a distributed and multistage beamforming architecture with station beam generated by 256 antennas and tied-array beam summed from multiple (up to 512) stations [6]. Station beams are formed by daisy-chaining 16 Tile Processing Modules (TPMs), where the pointing coefficients are calculated in advance for all antenna locations and converted to per-channel phase terms in the TPM beamformer [10]. This data independent beamformer is also called Classical BeamFormer (CBF), which is in a classical sense approximating a desired response of unity at a point of direction and zero elsewhere, or the desired response of known strong RFI source direction is zero. CBF is stable and resource-efficient, but can’t deal with the unknown interference. RFI from the sidelobes of station beam will corrupt the data and increase the effective bit-widths of data stream.

Spatial filtering method is particularly useful for array instruments and multi-feed single-dish radio telescopes. A variety of specific spatial filtering algorithms including maximum SNR, subspace projection, and multiple sidelobe cancelling have been studied for application to radio astronomical observing [11,12,13,14]. Most of these algorithms are based on short-term correlation matrices (the astronomical “visibilities”), which are preferred to be done on post-correlation data. The first demonstrate system with real-time spatial filtering RFI mitigation function is firstly used on the Phased Array Feed (PAF) receiver for Green Bank telescope [15]. The demonstration system consists of a 16 element phased array and carries out the real-time beamforming in HPCs. Except on Phased Array Feed (PAF), it’s rare to see research that applies these algorithms to pre-correlation process stage of radio astronomy aperture array. However, spatial filtering is widely used as adaptive beamformer in radar and communication systems [16,17,18,19].

Two critical reasons next generation phased array radio telescopes do not use an adaptive beamforming strategy are the distributed beamforming architecture and computational resource limitation. Take SKA-low as an example, signals from antennas are individually processed on different TPMs and partially summed sequentially, and no real-time data that aggregates signals form all the antennas is available, which makes it difficult to apply adaptive algorithm directly. Furthermore, most hardware resources (logic slices, block RAM and DSP48s) in the TPM are used by channelizer, which are almost three times of that used by beamformer [10], so adaptive beamforming algorithms with high computation complexity are not applicable.

The minimum variance distortionless response (MVDR) criterion is an efficient beamforming strategy, which only requires a priori knowledge of the steering vector of SOI. Classical algorithm sample matrix inversion (SMI) can directly obtain MVDR beamformer, but it needs to invert the sample covariance matrix, which is computational intense and is an undesirable operation in large array systems. The least mean squares (LMS) and the recursive least squares (RLS) approaches can avoid matrix inversion, however, the LMS type methods always suffer slow convergence rate and instability, while the RLS class algorithms have relative large computation burden [20]. Many adaptive beamforming techniques have been presented to alleviate the drawbacks of heavy computational burden and data transmission [21,22,23,24]. A widely considered one is the partially adaptive processing technique, which utilizes a fraction of the available adaptive dimensions of an array for adaptation [25]. However, this technique suffers from the inevitable degradation in array performance due to the fact that the degrees of freedom for beamforming are reduced. Another class of techniques are based on subarray processing, which divides an adaptive array into several subarrays [26,27]. The subarrays adjusts its own adaptive weights independently. However, the subarray processor can achieve similar array performance with the original array beamformer only if all the interference lie outside the mainlobe of the smallest sub-array, which limits the number of antennas in the smallest subarray. A Subarray linearly constrained minimum variance (SLCMV) algorithm, which is not limited by the smallest subarray, has been presented [24,28]. It does not suffer any loss in the Degree Of Freedom (DOF), and can operate parallelly on more than one operational modules with only a small amount of data to be exchanged between different modules. A sub-array MVDR (SAMVDR) beamformer based on sub-array optimization can reach a fast covergence rate and a lower dimension of matrix computation [29]. But the computational complexity of SAMVDR is in the order of $O(N^{2})$, which is much higher than SLCMV (O(N))).

In this paper, we proposed a real-time adaptive station beamforming strategy for SKA-low, which uses an efficient Parallel Least Mean Square (PLMS) algorithm based on the Least Mean Square (LMS) implementation of Linear Constrained Minimum Power (LCMP) beamformer. The proposed PLMS beamformer has deeply nulling performance to sidelobe RFI, as well as good mainlobe response similar to classical beamformer. It is restructured to adapt the radio astronomy scenarios, and has even fewer inter-module data exchanges than SLCMV algorithm, which is suitable for application on SKA-low. Comparing to the station beamformer currently in use, the proposed station beamforming strategy doesn’t require hardware redesign, but only adds four complex multiplications and two additions for each beamformer in the firmware, as well as a data feedback loop for beam data transfer (from the last TPM to the other TPMs).

The rest of this article is organized as follows. In Section 2, the signal model and adaptive beamforming strategy for phased array radio telescope are introduced. Then, the station layout and signal processing architecture of SKA-low telescope are introduced in Section 3, including both station beamformers currently in use and that we proposed. Section 4 presents the simulation results of beam pattern and output SINR, as well as the experimental results of FPGA implementation demonstrations. Finally, Section 5 concludes this article.

The following notation are used in this paper. Matrices and vectors are presented by boldface characters. The superscripts *, *^T^* and *^H^* denote complex conjugation, transposition, and Hermitian transposition, respectively. I is an identity matrix.

## 2. Adaptive Beamforming Strategy

### 2.1. Data Model

The next generation phased array radio telescope such as SKA-low aperture array uses a frequency domain beamforming architecture as shown in Figure 1 [30]. Data streams from the individual antennas are channelized and processed as narrowband signals in each frequency bin, which can be delayed by applying a dynamic phase correction to each individual channel.

Consider an aperture array made of *N* antenna elements, the array output vector at time *t* is $x\left(t\right)={\left[x_{1}\left(t\right),...,x_{N}\left(t\right)\right]}^{T}$. The total observation interval *T* is divided into K disjoint intervals of length ΔT and indexed with *k*. Channelizer converts the divided input vector from time domain to frequency domain, to generate a set of complex vectors. The complex vector $X_{\Delta T}\left(\omega_{m},k\right)$ is referred as frequency-domain snapshot, which is an N-dimensional vector corresponding to the Fourier series coefficient at $\omega_{m}$, where $\omega_{m}$ is the centre frequency of the *m* th frequency bin.

We assume that all the desired signals and interference can be modeled as plane waves, then the array output x(t) can be modeled as $x\left(t\right)=x_{s}\left(t\right)+n\left(t\right)$, where $x_{s}\left(t\right)$ is the received signal from various kinds of sources: astronomical source assumed as a stationary Gaussian random process, or man-made RFI assumed as either a deterministic signal or a Gaussian random process; n(t) is the noise assumed as a zero-mean Gaussian random process. Then,
(1)$$\begin{array}{cc} \; & X_{\Delta T}\left(\omega_{m},k\right)=X_{s,\Delta T}\left(\omega_{m},k\right)+N_{\Delta T}\left(\omega_{m},k\right). \end{array}$$

In the narrowband beamformer, $X_{\Delta T}\left(\omega_{m},k\right)$ is processed with a matrix filter $w^{H}\left(\omega_{m}\right)$ to form the output beam
(2)$$\begin{array}{c} Y_{\Delta T}\left(\omega_{m},k\right)=w^{H}\left(\omega_{m}\right)X_{\Delta T}\left(\omega_{m},k\right)=\sum_{i=0}^{N-1}w^{*}\left(\omega_{m}\right)X\left(\omega_{m},k:p_{i}\right), \end{array}$$
where $p_{i}$ denotes the position of the *i*th element, and $Y_{\Delta T}\left(\omega_{m},k\right)$ is a complex scalar variable. The narrowband beamformers for each frequency bin are uncoupled, therefore it is convenient to suppress the *m* subscript of ω. Assuming the spatial spectral matrix is known and is the same for each snapshot, X(ω) will be used to represent $X_{\Delta T}\left(\omega_{m},k\right)$ in one snapshot to keep the notation as simple as possible.

Consider a signal propagating along the direction a with a temporal frequency ω, the array manifold vector v(k) can be defined by the wavenumber k and $p_{i}$, where k=(ω/c)a and *c* is the speed of light. The array manifold vector incorporate all the spatial characteristics of the array.
(3)$$\begin{array}{c} v\left(k\right)={\left[e^{-jk^{T}p_{0}},e^{-jk^{T}p_{1}},\cdots ,e^{-jk^{T}p_{N-1}}\right]}^{T}. \end{array}$$

Then, the array output signal X(ω) can be correlated to the source signal F(ω) by the array manifold vector v(ω:k),
(4)X(ω)=F(ω)v(ω:k).

The weights in a data independent beamformer are designed so the beamformer response approximates a desired response independent of the array data or data statistics. Consider a signal arriving from a known direction $a_{s}$, a common choice for CBF weight vector is the array manifold vector $v(\omega :k_{s})$ with a normalization factor 1/N,
(5)$$w_{CBF}\left(\omega \right)=v\left(\omega :k_{s}\right)/N.$$
Then, the output beam of CBF can be written as
(6)$$Y_{CBF}\left(\omega \right)=w_{CBF}^{H}\left(\omega \right)X\left(\omega \right)=v^{H}\left(\omega :k_{s}\right)X\left(\omega \right)/N.$$

### 2.2. Linear Constrained Minimum Power Beamformer

Additional assumptions can be made on the aperture array data model:The astronomical signal usually has a signal-to-noise ratio (SNR) of −20 dB or less, so $S_{c}+S_{n}\approx S_{n}$, where $S_{c}$ and $S_{n}$ are the spectral matrices of astronomical source and noise, respectively [31];Harmful strong interference may has a interference-to-noise ratio (INR) range from 0 dB up to 40 dB, which is therefore much stronger than astronomical signal;$S_{n}$ are initially unknown, because SKA-low antenna element has a wide-open field of view and is susceptible to a variety of RFIs;

Thus for a radio astronomy aperture array the array, the output signal is dominated by noise and interference (if present). In addition, assuming that we know (or can measure) the statistics of the total received waveform x(t), but do not know the statistics of the signal and noise components. Then we can minimize the total output power subject to specific constraints to obtain an optimum beamformer.

Consider $M_{c}$ linear constraints, the constraint equation can be written as
(7)$$w^{H}C=g^{H},$$
where $w^{H}$ is 1×N, C is $N\times M_{c}$, $g^{H}$ is $1\times M_{c}$. We require that the columns of C be linearly independent, the first column of C is $v(\omega :k_{s})$ and the first element of g is 1, so that the processor is distortionless. We minimize the output power subject to the constraint in (Equation 7) to derive the Linear Constrained Minimum Power (LCMP) beamformer
(8)$$\min_{w}w^{H}S_{x}w,\;s.t.\;w^{H}C=g^{H},$$
where $S_{x}$ is the spectral matrix of the total received signal. This problem can be solved by using a Lagrange multiplier λ, the function to be minimized is
(9)$$J≜w^{H}S_{x}w+\left[w^{H}C-g^{H}\right]\lambda +\lambda^{H}\left[C^{H}w-g\right],$$
where λ is an $M_{c}\times 1$ vector. Taking the complex gradient of *J* with respect to $w^{H}$ and setting it equal to zero. Solving for $\lambda^{H}$ to get the optimum constrained processor
(10)$$w_{lcmp}^{H}=g^{H}{\left[C^{H}S_{x}^{-1}C\right]}^{-1}C^{H}S_{x}^{-1}.$$

If the noise is white and there is no RFI, then (Equation 10) reduces to
(11)$$w_{q}^{H}=g^{H}{\left[C^{H}C\right]}^{-1}C^{H},$$
which is independent of the array output data and called quiescent weight vector.

### 2.3. Least Mean Square Implementation

In actual applications, we must estimate the spatial spectral matrix ($S_{x}$) (or appropriate surrogates) from the incoming data. The resulting beamformers will adapt to the incoming data and referred to as adaptive beamformers. Three adaptive algorithms are most widely used: Sample Matrix Inversion (SMI) algorithm, Recursive Least Squares (RLS) algorithm and Least Mean Square (LMS) algorithm [20].

The SMI algorithm is a block processing algorithm in which the estimated spatial spectral matrix is substituted for the ensemble spectral matrix, that is effective in many applications. The disadvantage of the SMI approach is the computational complexity, in the order of $O(N^{3})$. The RLS algorithm potentially has performance that is similar to the SMI algorithm, but has a lower computational complexity of $O(N^{2})$. The LMS algorithm requires the least amount of computation (O(N)), which has the capabilities for real-time implementation on SKA-low station.

Frost present a steepest descent algorithm to find the LCMP beamformer [32], we will follow this reference and derive a parallel version for SKA-low implementation.

From (Equation 9), we want to minimize J, the gradient with respect to $w^{H}$ is
(12)$$\nabla_{w^{H}}=S_{x}w+C\lambda .$$

We define
(13)$$w\left(K\right)=w\left(K-1\right)+\alpha \left(-\nabla_{w^{H}}\right),$$
where α is a real parameter, which we refer to as the step size parameter. Then,
(14)$$\begin{array}{c} w\left(K\right)=w\left(K-1\right)-\alpha \nabla_{w^{H}}\left(J\left(K-1\right)\right)=w\left(K-1\right)-\alpha \left[S_{x}w\left(K-1\right)+C\lambda \left(K-1\right)\right], \end{array}$$
w(K) is required to satisfy the constraint (Equation 7). Solving for λ(K−1) and substituting into (Equation 14) gives
(15)$$w\left(K\right)=P_{c}^{⊥}\left[w\left(K-1\right)-\alpha S_{x}w\left(K-1\right)\right]+w_{q}$$
with $P_{c}^{⊥}=\left[I-C{\left(C^{H}C\right)}^{-1}C^{H}\right]$ and $w\left(0\right)=w_{q}$.

A simple estimate of $S_{x}$ is
(16)$${\hat{S}}_{x}=X\left(K\right)X^{H}\left(K\right).$$

In other words, the instantaneous values are used as estimates. Using (Equation 16) in (Equation 15) gives
(17)$$\hat{w}\left(K\right)=P_{c}^{⊥}\left[\hat{w}\left(K-1\right)-\alpha X\left(K\right){\tilde{Y}}_{p}^{*}\left(K\right)\right]+w_{q},$$
where
(18)$${\tilde{Y}}_{p}\left(K\right)={\hat{w}}^{H}\left(K-1\right)X\left(K\right).$$
${\tilde{Y}}_{p}\left(K\right)$ is the adaptive beamformer output of Kth snapshots, which is a scalar, then $\alpha X\left(K\right){\tilde{Y}}_{p}^{*}\left(K\right)$ is a N×1 vector. $w_{q}$ is the N×1 quiescent weight vector that can be pre-calculated. $P_{c}^{⊥}$ is a N×N vector that can be also pre-calculated.

Matrix multiplication operations $P_{c}^{⊥}\hat{w}\left(K-1\right)$ and $P_{c}^{⊥}X\left(K\right)$ are required for solving $\hat{w}\left(K\right)$, so that the complete $\hat{w}\left(K-1\right)$ and X(K) vectors are required. As mentioned in Section 1, the sample snapshots of the SKA-low station are distributed over multiple TPMs, thus no complete X(K) is available. (Equation 17) cannot be used directly on SKA-low station beamformer.

### 2.4. Parallel Least Mean Square Algorithm

To derive a parallel algorithm applicable to SKA-low station beamformer, we can simplify the constraints. The main constraint for SKA-low station beamformer is distortionless criterion, which can be written as $w^{H}v_{s}=1$. Consider this only constraint, then $C=v_{s}$, g=1,
(19)$$w_{q}=v_{s}{\left(v_{s}^{H}v_{s}\right)}^{-1}=v_{s}/N,\;P_{c}^{⊥}=I-w_{q}v_{s}^{H}.$$
(Equation 17) can be written as
(20)$$\begin{array}{cc} \hat{w}\left(K\right) & =\left(I-w_{q}v_{s}^{H}\right)\left[\hat{w}\left(K-1\right)-\alpha X\left(K\right){\tilde{Y}}_{p}^{*}\left(K\right)\right]+w_{q}\\ \; & =\hat{w}\left(K-1\right)-w_{q}v_{s}^{H}\hat{w}\left(K-1\right)-\alpha X\left(K\right){\tilde{Y}}_{p}^{*}\left(K\right)+\alpha w_{q}v_{s}^{H}X\left(K\right){\tilde{Y}}_{p}^{*}\left(K\right)+w_{q} \end{array}$$

According to the distortionless criterion, $v_{s}^{H}\hat{w}\left(K-1\right)={\left[\hat{w}{\left(K-1\right)}^{H}v_{s}\right]}^{H}=1$. Then
(21)$$\begin{array}{cc} \hat{w}\left(K\right) & =\hat{w}\left(K-1\right)-\alpha X\left(K\right){\tilde{Y}}_{p}^{*}\left(K\right)+\alpha w_{q}v_{s}^{H}X\left(K\right){\tilde{Y}}_{p}^{*}\left(K\right)\\ \; & =\hat{w}\left(K-1\right)-\alpha X\left(K\right){\tilde{Y}}_{p}^{*}\left(K\right)+\alpha v_{s}{\tilde{Y}}_{c}\left(K\right){\tilde{Y}}_{p}^{*}\left(K\right) \end{array}$$
where ${\tilde{Y}}_{c}\left(K\right)$ is the classical beamformer output
(22)$${\tilde{Y}}_{c}\left(K\right)={\hat{w}}_{q}^{H}X\left(K\right).$$
α, ${\tilde{Y}}_{c}\left(K\right)$ and ${\tilde{Y}}_{p}^{*}\left(K\right)$ are all scalars.

Assuming the station array with N antenna elements is evenly divided into M subarrays, the snapshot vector is partitioned into $X\left(K\right)={\left[X_{1}^{T}\left(K\right),X_{2}^{T}\left(K\right),\ldots ,X_{M}^{T}\left(K\right)\right]}^{T}$, where $X_{i}^{T}\left(K\right)$ is an $N_{i}\times 1$ vector and $\sum_{i=1}^{M}N_{i}=N$, $N_{i}$ is the antenna element number of subarray. w(K) and $v_{s}$ are correspondingly partitioned as
(23)$$\begin{array}{cc} \; & w\left(K\right)={\left[w_{1}^{T}\left(K\right),w_{2}^{T}\left(K\right),\ldots ,w_{M}^{T}\left(K\right)\right]}^{T}\\ \; & v_{s}={\left[v_{s1}^{T},v_{s2}^{T},\ldots ,v_{sM}^{T}\right]}^{T} \end{array}$$

Then, w(K) can be derived from (Equation 21),
(24)$${\hat{w}}_{i}\left(K\right)={\hat{w}}_{i}\left(K-1\right)-\alpha X_{i}\left(K\right){\tilde{Y}}_{p}^{*}\left(K\right)+\alpha v_{si}{\tilde{Y}}_{c}\left(K\right){\tilde{Y}}_{p}^{*}\left(K\right),$$
where
(25)$$\begin{array}{cc} \; & {\tilde{Y}}_{c}\left(K\right)=\sum_{i=1}^{M}v_{si}^{H}X_{i}\left(K\right)/N,\;{\tilde{Y}}_{p}\left(K\right)=\sum_{i=1}^{M}{\hat{w}}_{i}^{H}\left(K-1\right)X_{i}\left(K\right). \end{array}$$

The proposed PLMS Algorithm 1 is summarized as follows.
**Algorithm 1** algorithm of PLMS beamformer**Input:** α, $v_{si}$, $X_{i}$**Output:** ${\hat{w}}_{i}$, ${\tilde{Y}}_{c}$, ${\tilde{Y}}_{p}$1:Initialization: ${\hat{w}}_{oi}=v_{si}$, ${\hat{w}}_{i}=0$, ${\tilde{Y}}_{c}=0$, ${\tilde{Y}}_{p}=0$,2:Iteration procedure:3:**while** 
true 
**do**4:   **for** i=1,2,...,M **do**5:     ${\hat{w}}_{i}={\hat{w}}_{oi}-\alpha X_{i}{\tilde{Y}}_{p}^{*}+\alpha v_{si}{\tilde{Y}}_{c}{\tilde{Y}}_{p}^{*}$,6:     ${\tilde{Y}}_{c}\left(i\right)=v_{si}^{H}X_{i}/N$,7:     ${\tilde{Y}}_{p}\left(i\right)={\hat{w}}_{oi}^{H}X_{i}$,8:     ${\hat{w}}_{oi}={\hat{w}}_{i}$,9:     **return** ${\hat{w}}_{i}$,10:  **end for**11:  ${\tilde{Y}}_{c}=\sum_{i=1}^{M}{\tilde{Y}}_{c}\left(i\right)$, ${\tilde{Y}}_{p}=\sum_{i=1}^{M}{\tilde{Y}}_{p}\left(i\right)$,12:  $\alpha {\tilde{Y}}_{p}^{*}$,13:  $\alpha {\tilde{Y}}_{c}{\tilde{Y}}_{p}^{*}$,14:  **return** ${\tilde{Y}}_{c}$, ${\tilde{Y}}_{p}$.15:**end while**

A variable step size α(K) can be used for better convergence performance [20],
(26)$$\alpha \left(K\right)=\frac{\gamma }{\beta +X^{H}\left(K\right)X\left(K\right)},\;\beta >0,\;0<\gamma <2,$$
where β and γ provide normalized weighting of $X^{H}\left(K\right)X\left(K\right)$, Normally, β will be close to 1, and the typical values of γ are 0.005<γ<0.05. To minimize real-time computation, $X^{H}\left(K\right)X\left(K\right)$ can be not updated for every snapshot, but approximated by estimate obtained from the individually collected samples.

### 2.5. Computational Complexity

The computational complexity of PLMS algorithm is measured by the number of complex multiplications (CMs) within one iteration and one snapshot (K=1). Consider a step size with fixed value, $\alpha {\tilde{Y}}_{p}^{*}$ needs 1 CM and $\alpha {\tilde{Y}}_{c}{\tilde{Y}}_{p}^{*}$ needs another 1 CM before the computation of ${\hat{w}}_{i}\left(K\right)$. Then, for each element, ${\hat{w}}_{i}\left(K\right)$ needs 2 CMs, ${\tilde{Y}}_{c}$ needs 1 CM and ${\tilde{Y}}_{p}$ need 1 CM. Therefore, the computational complexity of PLMS is 4N+2, which is less than that of SLCMV algorithm (5N+3) [28]. Moreover, in SKA-low station, ${\hat{w}}_{i}\left(K\right)$, ${\tilde{Y}}_{c}$ and ${\tilde{Y}}_{p}$ can be computed parallelly over *M* TPMs, the computational complexity for each TPM is approximately 4N/M.

## 3. Description of the SKA-Low Station

### 3.1. Station Layout

The SKA1-low aperture array is composed of 131,072 dualpolarized antennas grouped in 512 stations of 256 antennas, each station has a diameter of about 40 m [6]. The antenna positions within each circular Station are randomized to avoid grating lobes, sidelobe structure, and scan-blindness effects.

Figure 2 shows an example of a quasi-random configuration for a 42 m station with 256 antennas, the antennas are divided into 16 tiles (denoted in different colors). The average spacing between elements is 2.1142 m, with a minimum spacing of 2 m, whilst most are concentrated between 2 m and 2.3 m. The simulation results in this paper are based on this configuration. This configuration may differ from the actual final configuration, but the relevant conclusions in this article are not affected.

### 3.2. Station Signal Processing

Signal processing inside a station includes coarse channelization, calibration and beamforming, which is performed in groups (tiles) of 16 antennas in 16 TPMs. TPMs are connected together in a flexible way using the 40 Gb high speed network, with non-blocking network switches. The block diagram of the station signal processing architecture is shown in Figure 3.

In the current signal processing design, station beamforming is performed by daisy-chaining the TPMs, as shown in black lines in Figure 3. Each ADC generates 800 Mbyte/s of 8 bit samples, and the channelizer selects a bandwidth of 300 MHz, oversampled to 356 Msample/s, with resolution increased to 12 + 12 bit complex samples. The aggregated data then reduced by a factor of 16 (one beam every 16 antennas, 17 Gbps) in the tile beamformer. The traveling sum on the 40 Gbps interface uses 16 + 16 bit complex samples, for a data rate, including packet overhead, of 23.2 Gbps. The traveling sum is shared by 384 frequency bins, then the data rate of sum beam for one frequency bin is about 62 Mbps. The output final beamformed data is represented with 8+8 complex samples, with a data rate of 11.6 Gbps.

To apply the proposed PLMS adaptive beamforming strategy on SKA-low station, we simply add one additional data loop to each TPM, as shown in red lines in Figure 3. Each additional data loop is used to transfer two sum beam data-classical beam and adaptive beam, at a data rate of 124 Mbps per frequency bin. One GTH transceiver on the FPGA has a bandwidth up to 13 Gbps, which can support about 100 frequency bins of loop data. Since RFI exists in only few fequency bins, one additional GTH transceiver is considered sufficient to enable the adaptive beamforming strategy. In addition, the PLMS adaptive beamformer outputs both classical and adaptive beams, allowing free selection of desired output.

## 4. Numerical and Experimental Results

### 4.1. Simulation Details

The simulations in this section are all based on the station layout as shown in Figure 2. The array is constrained to pass signals from broadside ($\left(\theta ,\phi \right)=\left(0^{\circ },0^{\circ }\right)$) with unit gain and all antennas have a same response. We assume that an additive white Gaussian noise is contained in the received data. The desired signal with signal-to-noise ratio (SNR) = −20 dB is located on the array’s broadside. Two incoherent interference with interference-to-noise ratio (INR) = 0, 10 and 20 dB are incident from directions of (θ,ϕ)=$\left(13.5^{\circ },0^{\circ }\right)$ and $\left(41^{\circ },0^{\circ }\right)$, respectively. Three simulation frequencies of 50, 200 and 350 MHz are selected to cover the entire operating band. The initial values $w_{i}$ are set to $v_{si}$, for i=1,2,...,16. The initial value of step size α are set to $5\times 10^{-6}$. In all cases we only examine steady-state performance. The simulation results presented for each example are obtained after averaging the results of 100 independent runs.

The CBF and diagonal loading SMI beamformers are also simulated for comparison, where $w_{CBF}^{H}=v_{s}^{H}/N$, and
(27)$$w_{SMI}^{H}=\frac{v_{s}^{H}{\left({\hat{S}}_{x}+\sigma_{L}^{2}I\right)}^{-1}}{v_{s}^{H}{\left({\hat{S}}_{x}+\sigma_{L}^{2}I\right)}^{-1}v_{s}},$$
$\sigma_{L}^{2}$ is the fixed loading value. The load-to-white noise level is defined as $LNR=\sigma_{L}^{2}/\sigma_{w}^{2}$, with a default value of 0 dB.

### 4.2. Performance Metrics

Prior to introducing our simulation results, it is instructive to establish performance metrics for evaluating the relative performances of the proposed approach. Beam pattern is the frequency-wavenumber response function evaluated versus the direction, which is a critical performance metric of the station array. The optimum array pattern has a perfect null at the direction of the interference, but for finite INR, it creates a partial null (or notch) whose depth is adjusted to minimize the output power.

The straightforward way to compute the station beam is the simple geometric-based Array Factor (AF) approach, as used in traditional array analysis and design. The AF is given by
(28)$$AF\left(k\right)=w^{H}v\left(k\right),$$
therefore, the array beam pattern is the isolated pattern of antenna element multiplied by AF,
(29)BP(k)=EP(k)AF(k).

This simple AF approach has limited accuracy, primarily because it ignores both mutual coupling and the variability in element patterns. A more accurate model of the station, which takes into account the antenna interactions, is given by computing a different element pattern for each antenna [33].

In this research, we focus on the performance of different adaptive beamforming strategies, based on a fixed array layout. For simplicity, all element are assumed to have a same response of EP(k)=1, then BP(k)=AF(k). The following simulations in this article are all based on this assumption.

Another critical performance metric is the output signal-to-noise ratio (SNR), $SNR_{o}≜P_{s}/P_{n}$, where the “noise” is assumed to contain both the white sensor noise and the interference. To emphasize that n contains both noise and interference, we use $SINR_{o}$ here.
(30)$$SINR_{o}≜\frac{\sigma_{s}^{2}{\left|{\hat{w}}^{H}\left(K\right)v_{s}\right|}^{2}}{{\hat{w}}^{H}\left(K\right)S_{n}\hat{w}\left(K\right)}.$$

We want to investigate the $SINR_{o}$ behavior versus number of snapshots *K*, input INR, and step size α.

### 4.3. Simulated Beam Patterns

Figure 4 shows the simulated beam patterns of the station with INR = 20 dB, *K* = 10,000 and $\alpha =5\times 10^{-6}$ at 50 MHz, 200 MHz and 350 MHz, respectively. The left part shows the top view of the 3D beam pattern versus $u_{x}$ and $u_{y}$ (defined in sine space: $u_{x}=\sin\;\theta \;\cos\;\phi ,u_{y}=\sin\;\theta \;\sin\;\phi$) on a logarithmic scale, and the right part shows the beam pattern cut on XOZ plane in (θ,ϕ)-space (spherical coordinate system). As one can see, all the beam patterns have deep nulls (lower than −60 dB) at the directions of two interferer.

In Figure 5, representative beam patterns of PLMS beamformer and CBF for various *K* are plotted. There are two striking features: (1) The PLMS beam pattern is changed iteratively from the CBF beam pattern as a starting point, meaning a same performance as the CBF when no interference is present. (2) Throughout the iterations, the beam pattern maintains a main lobe nearly identical to that of CBF, meaning that the array response to the observing space region is stable, which is of most interest to astronomers. These features make the PLMS beamforming strategy suitable for SKA-low aperture array.

Figure 6 shows the simulated beam patterns of PLMS, SMI and CBF in a same scenario as Figure 4. The final beam pattern obtained by PLMS beamformer is almost the same as that of the SMI beamformer. Furthermore, both of them have mainlobes that are very similar to CBF.

Figure 7 shows the simulated beam patterns of PLMS beamformer for various INR, with the same step size of $\alpha =5\times 10^{-6}$. In low INR scenarios, the PLMS beamformer needs more snapshots to converge. For example, 30 snapshots is enough to get a stable beam pattern when INR = 20 dB, but the snapshot number for convergence increases to more than 6000 in a INR = 0 dB scenario. This feature can be seen more explicitly by $SINR_{o}$ simulation results in the next section.

### 4.4. Simulated Output SINR

Figure 8 shows the simulated average $SINR_{o}$ vs. *K* at 50, 200 and 350 MHz, with the step size $\alpha =5\times 10^{-6}$, input INR varying from 0 to 20 dB in 10 dB steps. In general, when the step size α is constant, the most important factor affecting convergence speed is the value of input INR, the larger the INR the faster the convergence. When sharing the same INR, the average $SINR_{o}$ at different frequencies converge to similar values, and the convergence speed depends on the initial $SINR_{o}$, the larger the initial $SINR_{o}$ the fast the convergence. As one can see that, the beamformer has a similar final $SINR_{o}$ of about 4 dB when INR = 0 and 10 dB, but lower value of about 2.3 dB when INR = 20 dB.

Figure 9 shows the simulated $SINR_{o}$ vs. *K* at 50 MHz for different beamformers, with input INR = 0, 10 and 20 dB, $\alpha =5\times 10^{-6}$. The average $SINR_{o}$ of CBF gives a initial and iteration start point for PLMS beamformer. SMI beamformer has a optimal performance that less affected by input INR, which provides a limit value reference for PLMS performance optimization. As one can see, the average $SINR_{o}$ of PLMS for INR = 20 dB doesn’t reach the optimal value, and the number of snapshots required for convergence is surprisingly less than SMI. It is reasonable to suspect that the iterative approximation is not optimal.

Figure 10 shows the simulated $SINR_{o}$ vs. *K* at 50 MHz for INR = 20 dB, with different α=50,10,5,3.1 and $1\times 10^{-7}$. As can be seen from the figure, decreasing the step size improves the final $SINR_{o}$ of PLMS beamformer, and it almost reaches the optimal performance when $\alpha =1\times 10^{-7}$. However, smaller step size reduces the convergence speed, so there is a trade-off to be made here. According to (Equation 26), we can obtain an estimate of α based on the actual snapshots. In the simulation shown in Figure 10, when γ=0.01 and β=0.99, the step size $\alpha =3.1\times 10^{-7}$, then the final average $SINR_{o}$ = 3.87 dB, which differs from the optimal value by only 0.15 dB. However, the number of snapshots required for convergence is reduced to 600 in this configuration. In contrast, the number of required snapshots is 2000 when $\alpha =1\times 10^{-7}$, with a final average $SINR_{o}$ = 3.95 dB.

### 4.5. FPGA Implementation Demonstration

SKA-low is under construction and the initial six station array have not yet completed [6]. The proposed real-time adaptive station beamforming strategy is also applicable to other large-scale aperture arrays other than SKA-low. A digital beamforming system developed for large-scale aperture array radio telescope was considered well suited for the experimental validation of this tragegy [34]. However, small-scale functional test of the algorithm is necessary before the system can be put to use.

To estimate the hardware complexity of the proposed algorithm, we implemented PLMS, SMI, and CBF algorithms on a NI PXIe-7976R FPGA module. The FPGA implementation demonstrations can carry out the digital processing of 8-channel input 16 bit real signals, including integer and fractional sample delays, digital down-conversion (DDC) to generate complex signals (represented in In-phase and Quadrature components), and digital beamforming. Due to the limitation of FPGA hardware resources, there is no frequency channelizer and only one beamformer was implemented in each demonstration. The functional modules of CBF, SMI and PLMS beamformer can be used in ditial beamforming systems of future aperture arrays. All other functional modules except beamformer in different FPGA demonstrations are identical. Narrowband signals and white noise were used as input test signal to simulate the processing and beamforming behaviour in one single frequency bin.

The beamformers (PLMS, SMI and CBF) in FPGA demonstrations are based on (Equation 5), (Equation 27) and (Equation 24), respectively. They share the same complex weighting coefficients, which are represented as 16 bits signed fixed-point numeric (with 6 bits integer word length) I + 16 bits Q. The PLMS beamformer has an additional configurable parameter - step size α, which is a 24 bits unsigned fixed-point numeric with 1 bit integer word length. The diagonal loading value of SMI beamformer is not configurable, but is estimated from snapshots by $\sigma_{L}^{2}=std\left(diag\left({\hat{S}}_{x}\right)\right)$, where ${\hat{S}}_{x}=\left(\Sigma_{k=1}^{L}X^{H}\left(k\right)X\left(k\right)\right)/L$, L is set to 64 here. To reduce the computational complexity and improve real-time performance, Cholesky decomposition is used for solving the inverse of ${\hat{S}}_{x}+\sigma_{L}^{2}I$ in the SMI beamformer, which avoids division operation. The details of the implementation of the Cholesky decomposition are not presented here, but it’s still worth mentioning that floating point numeric is used in some intermediate operations to minimise the error.

Three FPGA implementation demonstrations have been coded in Labview, synthesis and implementation were performed on the NI PXIe-7976R FPGA module with the default clock frequency of 40 MHz. The FPGA used in this module is Kintex-7 XC7K410T, with 254,200 LUTs, 1540 DSP48 Slices (25 × 18 Multiplier) and 28,620 kbits Embedded Block RAM. The implementation results are summarized in Table 1, which show that the PLMS beamformer used 1102 more slices and 138 more DSP48s than CBF, while the SMI beamformer used 9956 more slices and 51 more DSP48s than CBF. The PLMS beamformer does not store and process intermediate matrices, so it uses much fewer slice than SMI.

The PLMS beamformer is a real-time processor, whose weight coefficients are updated with each snapshot. In contrast, the SMI beamformer is a block processor, whose operations such as solving average covariance matrix, Cholesky decomposition, and matrix multiplication take many clock cycles, and thus the DSP48s can be time-multiplexed; this is the reason why the SMI beamformer uses less DSP48s resources than the PLMS beamformer. The different real-time performance of PLMS and SMI can be easily demonstrated by experiment, as shown in Figure 11. Where SMI weighting coefficients are updated after more than 9000 snapshots, while PLMS weighting coefficients are updated in real time and converge within 100 snapshots. Since FPGA is real-time processor, its processing speed depends on the clock frequency and the amount of parallelism. Assuming a FPGA running at a clock frequency of 100 MHz (which is usually higher in practice) and quad-parallelism. The SMI beamformer requires about 2250 clock cycles to update the weighting coefficients once, or 22.5μs, while the PLMS beamformer is updated once per clock cycle, or 0.1μs. In comparison, the real-time beamformer using GPU takes ms order of time to complete one weight coefficient update [35]. The FPGA-based beamformer has an advantage in terms of real-time performance.

The waveforms shown in Figure 11 are in-phase components of the complex signals, and the input signal is one of the 8-channel input test signals. Test signal is a monochromatic tone (RFI) with a frequency of 100 MHz and amplitude of 256 ADC units RMS added to a white noise with a RMS level of 16 ADC units (INR≈ 24 dB), and then uses the input delay compensation to simulate a range of physical delays in the antenna signals, generated by a Labview host program. The referred ADC is the NI PIXe-5162 module working at a sample rate of 625 Msample/s. In this experiment, we consider a Uniform Linear Array (ULA) with eight elements with a inter-element spacing of 1.5 m, the steering angle is $0^{\circ }$, the RFI incident angle is $20^{\circ }$, the step size $\alpha =4.77\times 10^{-7}$. Weighting coefficients can be read from the FPGA demonstrations via host control program. Beam patterns are calculated from the obtained weighting coefficients and are shown in Figure 12, where operating frequency is 100 MHz and other parameter are same as the experiment configurations. Both SMI and PLMS beamformer create deep nulls in the direction of RFI and have similar main lobes.

Since the ULA in this experiment has only 8 elements, which is much less than the SKA-low station, the beam width of this ULA is much wider than that of the SKA-low station, and the DOF of this ULA is much less. Therefore the shape of the mainlobe is more susceptible to sidelobe RFIs in this ULA. Figure 13 shows the beam patterns of three beamformer with two RFIs are incident at $20^{\circ }$ and $40^{\circ }$, respectively. It can be seen that with the reduction of available degrees of freedom, the depth of nulls in the beam pattern of adaptive beamformers decreases, but still provides effective suppression of the RFIs. For the PLMS algorithm proposed in this paper, since only one distortion-less constraint is used, the available DOF is N−1, which, theoretically, can be used to suppress N−1 RFIs. However, in practice, when the number of RFIs approaches N−1, the suppression effect of the adaptive beamformer decreases rapidly. Since SKA-low uses a frequency domain beamformer, for M frequency bins, it can theoretically handle up to M*(N−1) RFIs. Taking into account the actual environment in which the system is located, while minimising the usage of the FPGA’s IO resources as much as possible, as described in Section 3.2, it is recommended that M does not exceed 100.

## 5. Conclusions

The SKA-low aperture array adopts a classical station beamformer, which can be easily implemented by serially accumulating partial beam data, but cannot deal with unknown RFI. A real-time adaptive beamforming strategy is proposed for SKA-low station, which uses an efficient PLMS algorithm derived from the LMS implementation of LCMP beamformer. The advancement of this strategy is that it can be used in the pre-correlation process stage of phased array telescopes, and its implementation is based on FPGAs. The proposed strategy has low computational complexity compared to commonly used adaptive beamforming methods, and can be processed in parallel. Beam pattern simulation results of PLMS algorithm applied on the SKA-low station show deep nulls at sidelobe RFI directions. SINR simulation results of the proposed algorithm shows the same performance as the classical LMS algorithm, which is close to the optimal performance with suitable step size configuration. Functional test of the PLMS algorithm was carried out on a NI PXIe-7976R FPGA module, and the test results were as expected, proving the validity of the method. Larger-scale experiments will be carried out in the future on a digital beamforming system developed specifically for phased array radio telescopes.

## Figures and Tables

**Figure 1 sensors-24-04723-f001:**
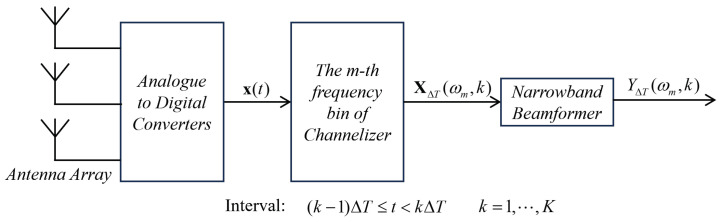
Frequency domain beamformer used in phased array radio telescope.

**Figure 2 sensors-24-04723-f002:**
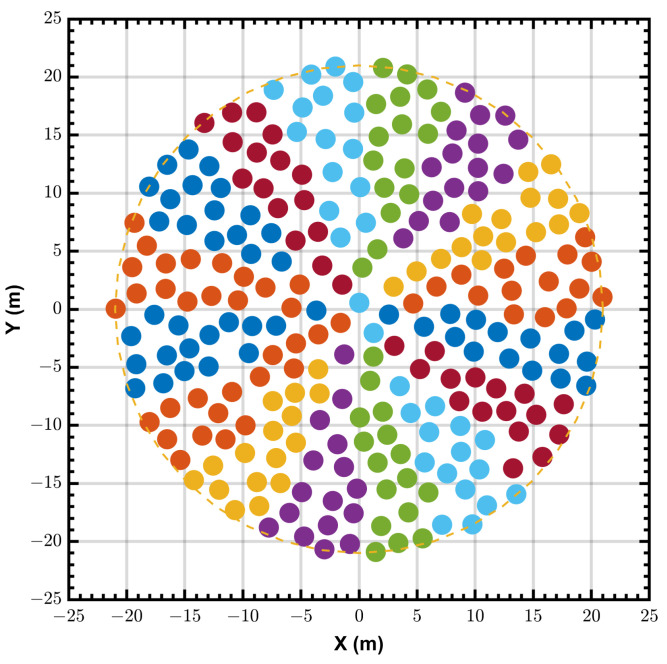
Quasi−random station antenna layout. 256 elements are divided into 16 tiles and each tile contains 16 antennas. Different tiles are denoted in different colors.

**Figure 3 sensors-24-04723-f003:**
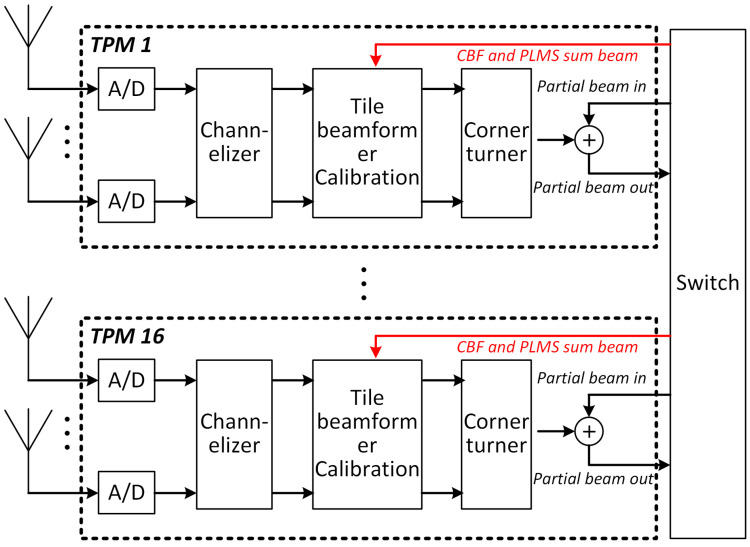
Signal processing block diagram of the SKA-low station. The black lines represent the current processing flow, while the red lines are the addition data loops to enable PLMS beamforming strategy.

**Figure 4 sensors-24-04723-f004:**
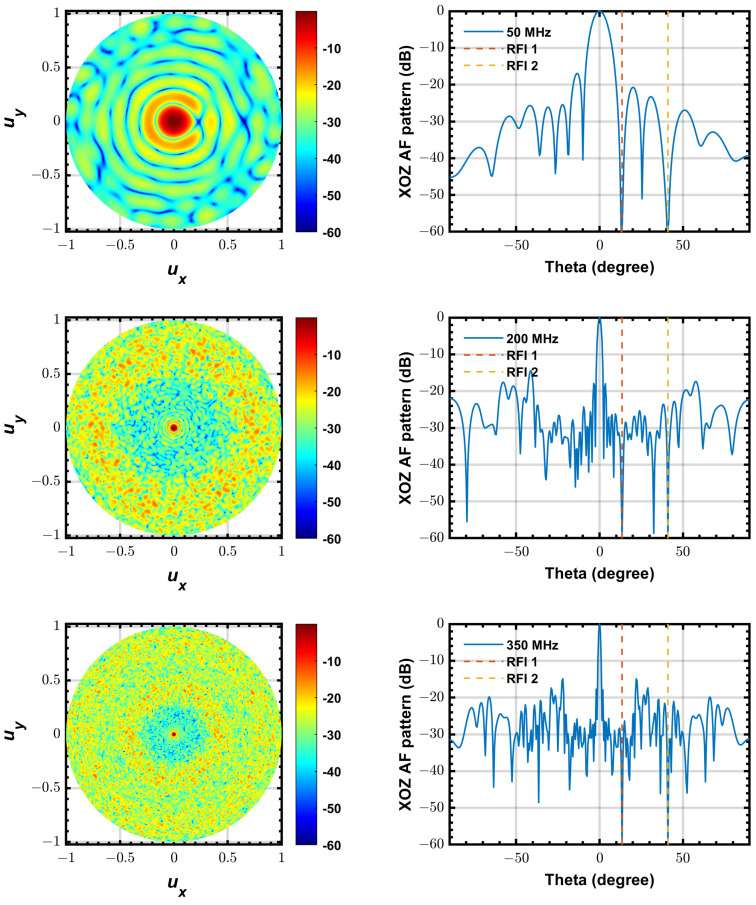
Beam pattern of the station with PLMS beamformer at 50, 200 and 350 MHz. (**left**) top view of the 3D pattern, (**right**) beam pattern cut on XOZ plane.

**Figure 5 sensors-24-04723-f005:**
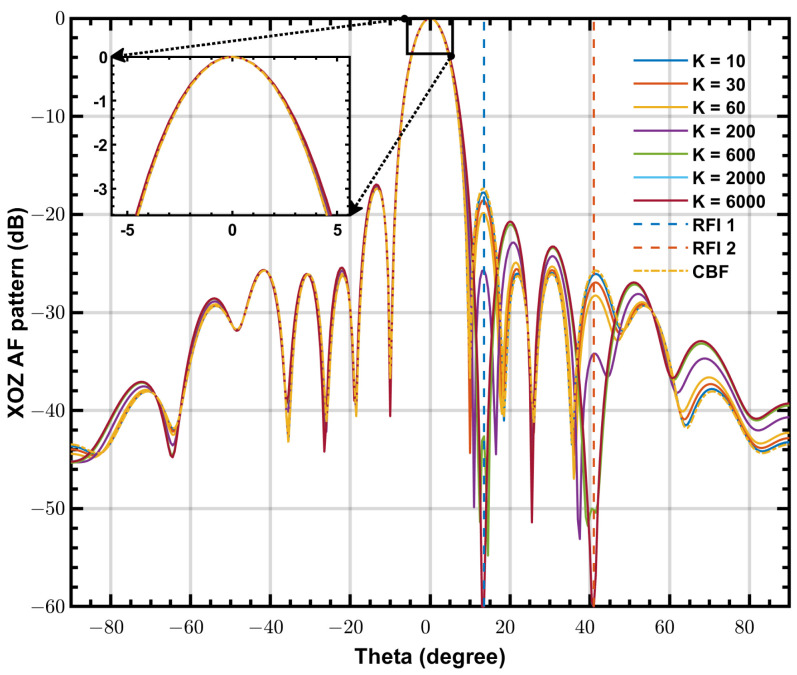
Representative beam patterns of PLMS beamformer and CBF for various K at 50 MHz.

**Figure 6 sensors-24-04723-f006:**
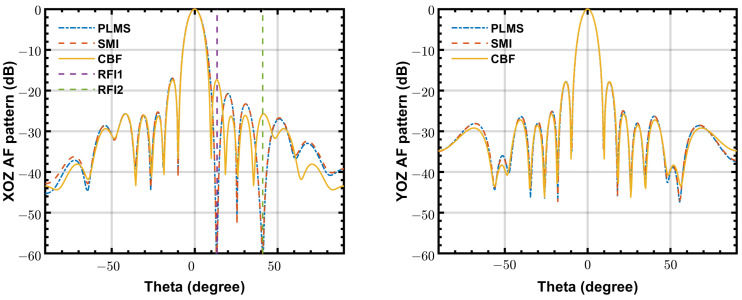
The beam patterns of PLMS, SMI and classical beamformer at 50 MHz, INR = 20 dB, K=10,000 and $\alpha =5\times 10^{-6}$. (**left**) beam pattern cut on XOZ plane, (**right**) beam pattern cut on YOZ plane.

**Figure 7 sensors-24-04723-f007:**
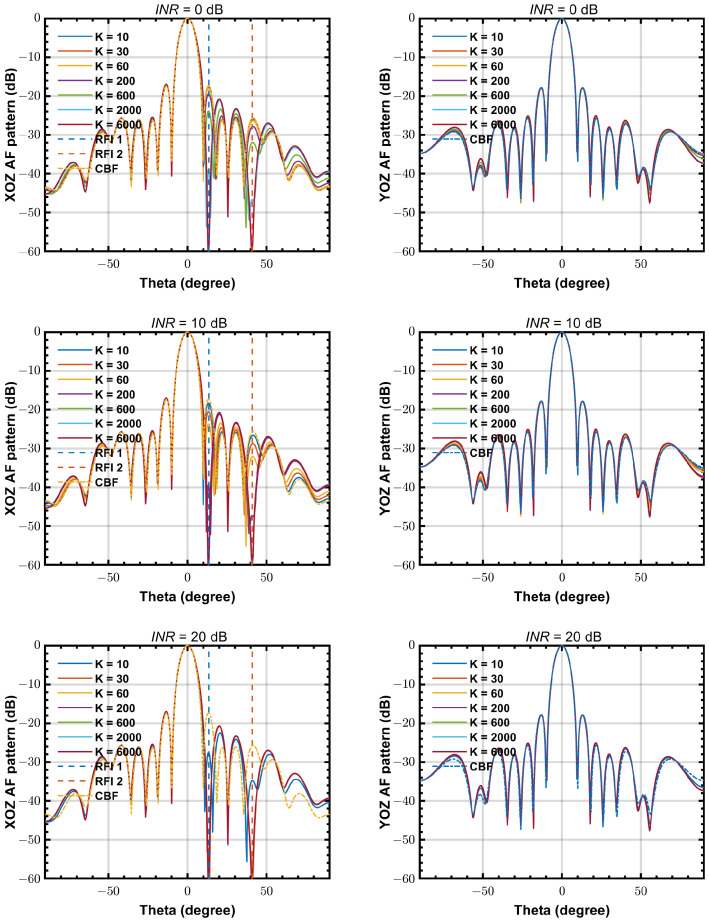
The beam patterns of PLMS beamformer at 50 MHz with different INR. (**left**) beam pattern cut on XOZ plane, (**right**) beam pattern cut on YOZ plane.

**Figure 8 sensors-24-04723-f008:**
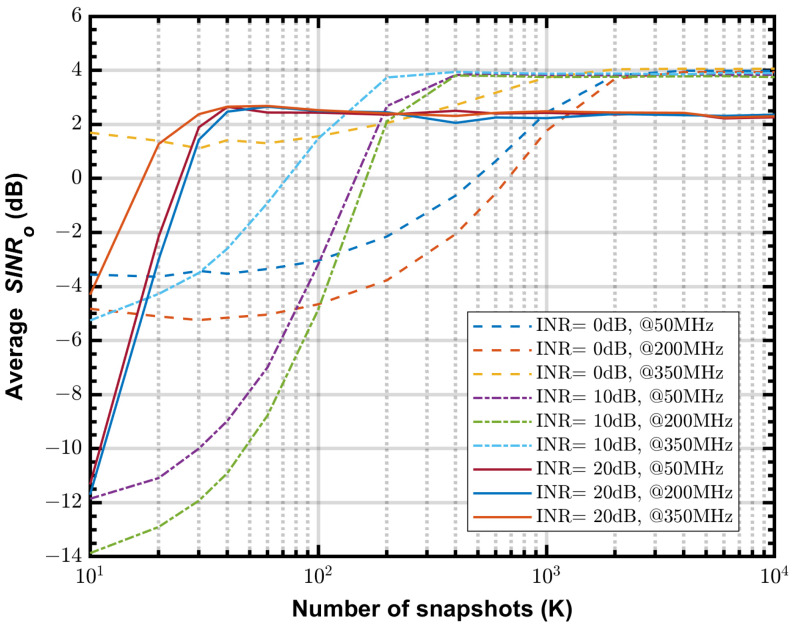
The average $SINR_{o}$ vs. *K* at 50 MHz, 200 MHz and 350 MHz, input INR = 0 dB, 10 dB, 20 dB, $\alpha =5\times 10^{-6}$.

**Figure 9 sensors-24-04723-f009:**
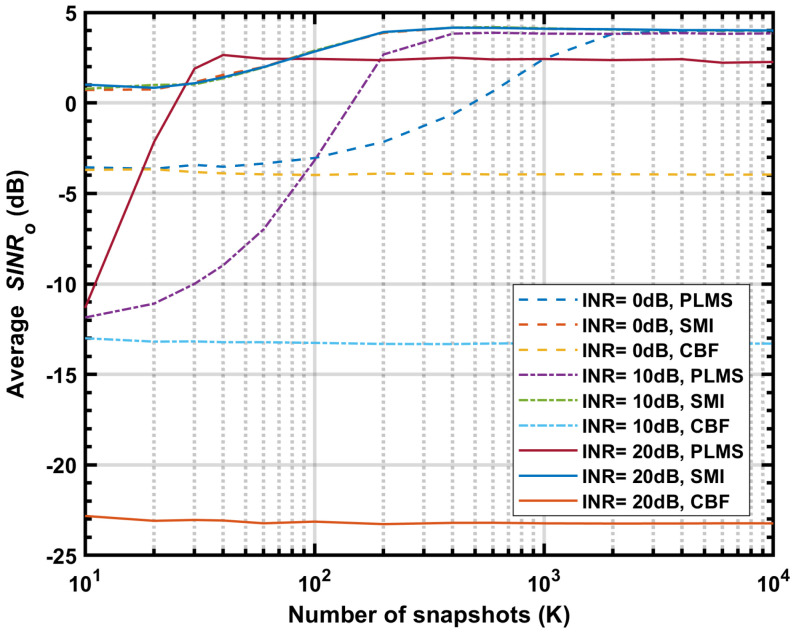
The average $SINR_{o}$ vs. *K* at 50 MHz for PLMS, SMI and CBF, INR = 0 dB, 10 dB and 20 dB, $\alpha =5\times 10^{-6}$.

**Figure 10 sensors-24-04723-f010:**
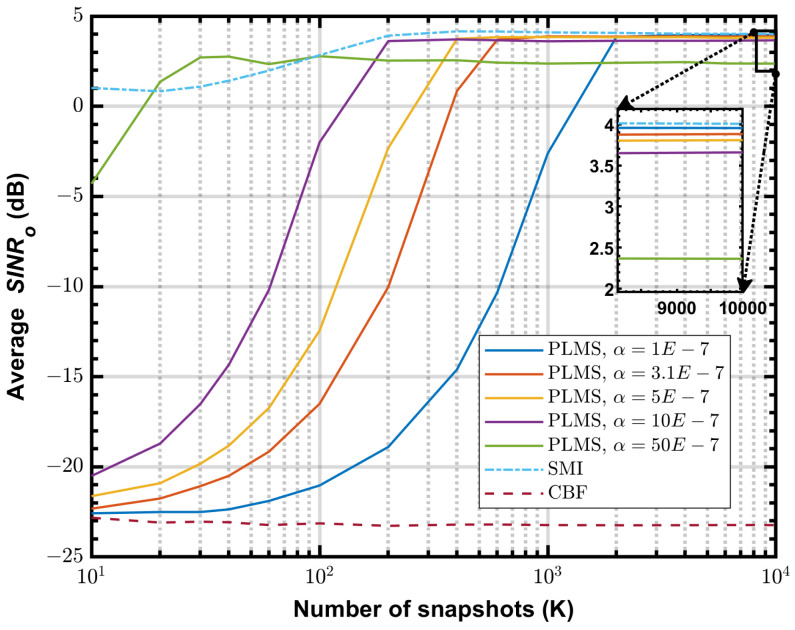
The average $SINR_{o}$ vs. *K* at 50 MHz, INR = 20 dB, α=50,10,5,3.1 and $1\times 10^{-7}$.

**Figure 11 sensors-24-04723-f011:**
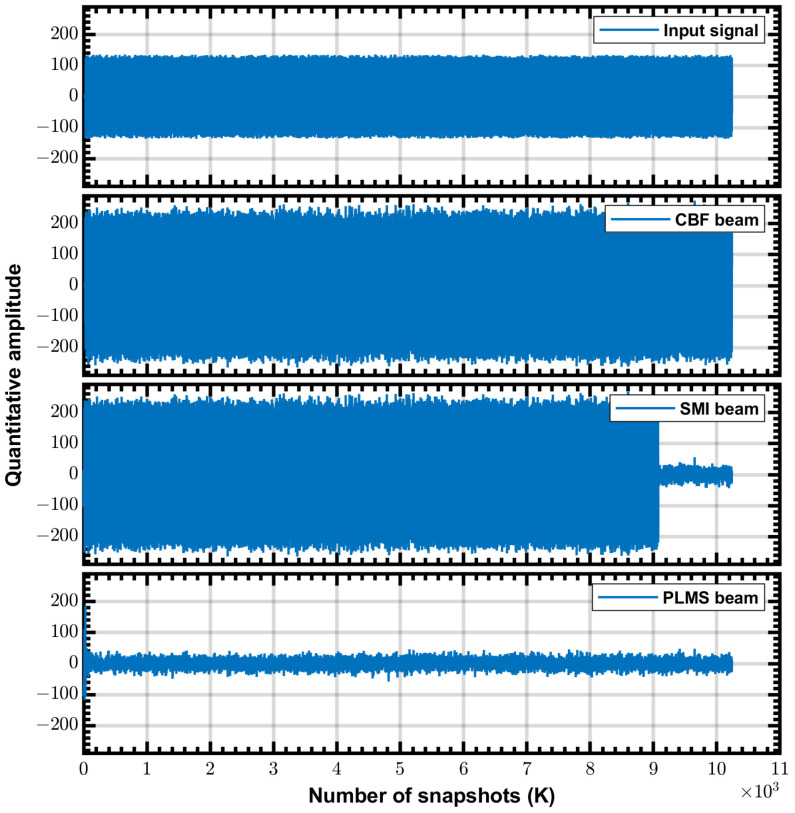
Test results of the three FPGA demonstrations, where in–phase component waveforms of the complex signals are shown.

**Figure 12 sensors-24-04723-f012:**
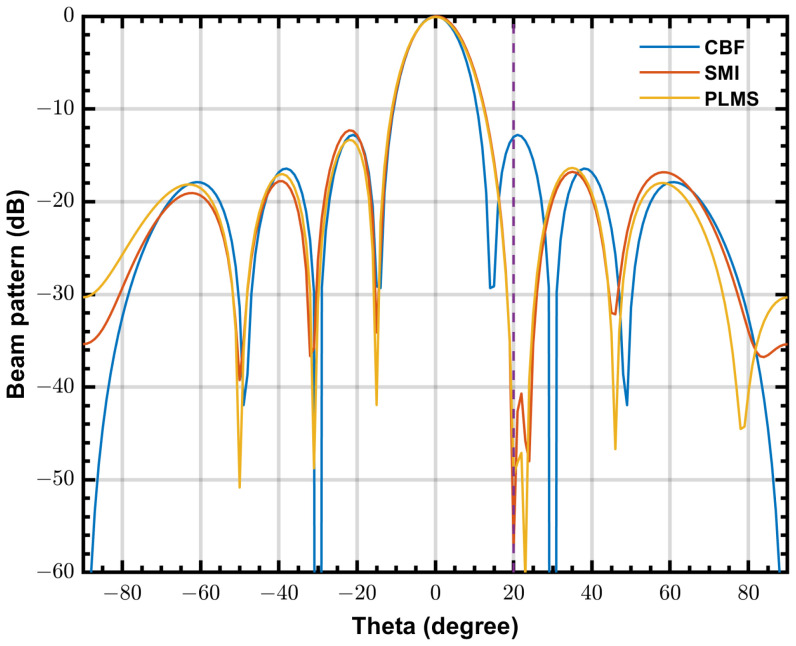
The beam patterns calculated from the final output weighting coefficients of three FPGA demonstrations. One RFI is incident at $20^{\circ }$.

**Figure 13 sensors-24-04723-f013:**
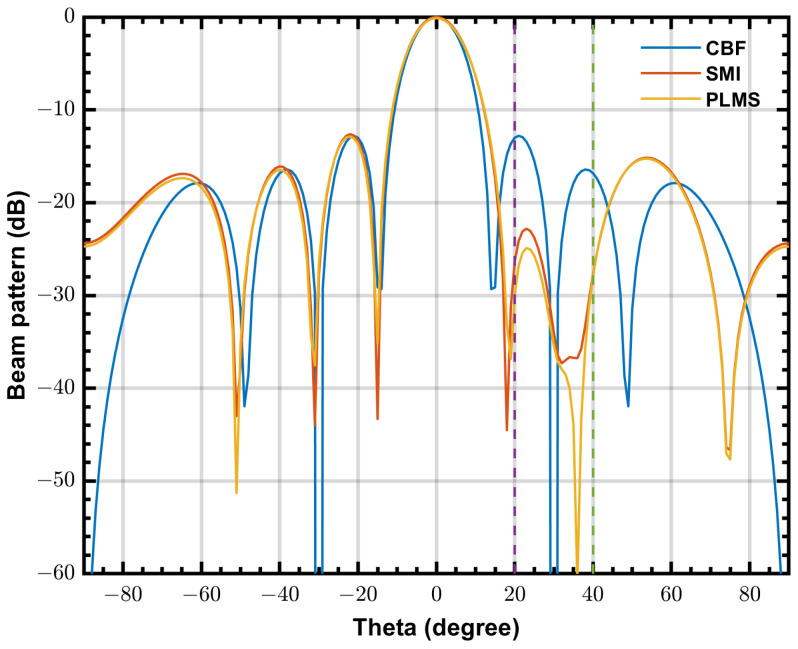
The beam patterns calculated from the final output weighting coefficients when two RFIs are incident at $20^{\circ }$ and $40^{\circ }$, respectively.

**Table 1 sensors-24-04723-t001:** Resource Utilization of Three FPGA Demonstrations.

Demo	Total Slice	Slice Register	Slice LUTs	Block RAMs	DSP48s
PLMS	28,887	80,543	76,861	147	826
SMI	37,741	110,287	103,082	200	739
CBF	27,785	80,088	71,565	147	688

## Data Availability

Data are contained within the article.

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
