# Peer review of "A Real-Time Adaptive Station Beamforming Strategy for Next Generation Phased Array Radio Telescopes"

_sensors, 2024, doi:10.3390/s24144723_

Round 1

Reviewer 1 Report

Comments and Suggestions for Authors

The manuscript is well written and contains interesting novel results. I'm sure it will be beneficial for readers who seek information about radio telescope. Some comments I would like to ask authors to consider are:

1. Did the parameters used in the equations need to be bold?

2. I feel like the results and discussion section mostly contains the repeat of what is already shown in the plots/figures. In my opinion, authors should add rationales for such findings. This will surely improve the quality of the manuscript.

3. Can authors include more recent references? I found that some statements need references for readers to find more information.

Comments on the Quality of English Language

Minor check is recommended.

Author Response

Comments 1:  Did the parameters used in the equations need to be bold?

Response 1: Thank you for pointing this out. Yes, the boldface characters denote matrices and vectors. Therefore, we have have added one paragraph to describe the notation are used in this paper at the end of chapter I (Introduction) .

Comments 2: I feel like the results and discussion section mostly contains the repeat of what is already shown in the plots/figures. In my opinion, authors should add rationales for such findings. This will surely improve the quality of the manuscript.

Response 2: Thank you for pointing this out. We agree with this comment. Therefore the Chapter 5 (conclusion) is reorganized.

Comments 3:  Can authors include more recent references? I found that some statements need references for readers to find more information.

Response 3: Thank you for pointing this out. We agree with this comment. Recent references have been added. And it should be noted that in Chapter 1, we have added descriptions about the first and only presented demonstrate system with real-time spatial filtering RFI mitigation function, as well as the reference. Additional descriptions about another similar sub-array adaptive beamforming algorithm name of SAMVDR  were also added in Chapter 1.  All the added references are highlighted in blue.

Reviewer 2 Report

Comments and Suggestions for Authors

The review report

The manuscript title: A Real-time Adaptive Station Beamforming Strategy for Next Generation Phased Array Radio Telescopes.

In this manuscript the authors propose a real-time adaptive beamforming strategy for the SKA-low station using the parallel least squares (PLMS) algorithm. Simulations of the beam pattern and SINR output were performed which showed very clear performance for sidelobe RFI, as well as good mainlobe response similar to a classical beamformer. The study also included determining the conditions for obtaining the best convergence performance.

The authors used references relevant to the topic of their article. The manuscript is appropriately prepared, the conclusions presented are consistent with the evidence, and arguments presented, and address the main question raised.

However, this article can be accepted for publication after taking into account the following points:

1. The study lacks comparison with actual experimental systems. Comparison with experimental results is a prerequisite for accepting the results of any theoretical study.

2. The research lacks actual comparisons with the results of similar previous research.

3. The number of references is insufficient and most of them are relatively old. In research that deals with modern technical topics, the reference study should be extensive and based on an important number of modern references.

4. There is a lot of information and equations that must be documented with references.

5. Reference 26 should be cited in the caption of Figure 1.

6. The explanation of Figure 12 is insufficient.

7. On line 343 the authors state that "the number of snapshots required for convergence 343 is reduced to 600 in this configuration. In contrast, the number of required snapshots is 344 2000 when α = 1 × 10−7". How does this reduction affect the final SINRo?

Author Response

Comments 1: The study lacks comparison with actual experimental systems. Comparison with experimental results is a prerequisite for accepting the results of any theoretical study.

Response 1: Thank you for pointing this out. We agree with this comment. Therefore A description has been added to the beginning of section 4.5.  As there is no fully operational SKA-low station yet, we consider future validation of our approach in a self-developed digital beamforming system, but a small-scale functional validation is necessary before a full experimental validation. In Section 4.5 of this paper, we introduced the functional test, where the beamformer functional module used here will be used in the full experimental system.

Comments 2: The research lacks actual comparisons with the results of similar previous research.

Response 2: Thank you for pointing this out. We agree with this comment. Since it's rare to see research that applies these algorithms to FPGA process of radio astronomy aperture array, the processing speed of real-time processor based on GPU is compared in Chapter 4.5. The FPGA-based beamformer has an advantage in terms of real-time performance. The revised content can be found on line 421, highlighted in blue.

Comments 3: The number of references is insufficient and most of them are relatively old. In research that deals with modern technical topics, the reference study should be extensive and based on an important number of modern references.

Response 3: Thank you for pointing this out. We agree with this comment. Reference about other phased array radio telescopes and the first demonstrate system with real-time spatial filtering processor based on GPU are added to Chapter 1. Some references related to the algorithm are also added. The revised content are highlighted in blue.

Comments 4. There is a lot of information and equations that must be documented with references.

Response 4: Thank you for pointing this out. We agree with this comment. Related references are added in the modified manuscript.

Comments 5. Reference 26 should be cited in the caption of Figure 1.

Response 5: Thank you for pointing this out. We agree with this comment. Therefore it has been modified, which can be found on line 92.

Comments 6. The explanation of Figure 12 is insufficient.

Response 6: Thank you for pointing this out. We agree with this comment. Therefor we added description of figure 12 and added another figure - figure 13, which can be found from line 434 to 456.

Comments 7. On line 343 the authors state that "the number of snapshots required for convergence 343 is reduced to 600 in this configuration. In contrast, the number of required snapshots is 344 2000 when α = 1 × 10−7". How does this reduction affect the final SINRo?

Response 7: Thank you for pointing this out. The SINRo = 3.95dB when α = 1 × 10−7. Related description is added to line 371. We added this information on line 373.

Reviewer 3 Report

Comments and Suggestions for Authors

Authors presented a real-time adaptive beamforming algorithm for suppressing sidelobe RFI's. They included detailed a detailed theory for the proposed algorithm. They implemented the proposed algorithm on a commercial FPGA module. The manuscript is well-written to the level of a high-ranking journal.

My comments are as follows.

1. Alpha, beta, and gamma in Equation (26) are presented without elaboration. Please add some explanation on Equation (26).

2. Authors presented a beamforming algorithm for an SKA-low station. Please explain how to extend the station-level algorithm to the 512 stations of the SKA1-low aperture array. Or rather explain how 512 stations work together.

3. The foremost merit of the proposed algorithm is the processing speed. Please consider adding a scale of processing time in Figures 8 to 9.

4. Please check if the title of Figure 12 is correct.

5. Figure 12:

1) An FPGA implementation of Figure 12 shows nulling for one angle while simulation results of Figures 5 to 7 show nulling at two angles. Please consider FPGA nulling at two angles.

2) The main beam change is appreciable in Figure 12 while it is negligible in Figures 5 to 7. Please add some comments.

6. With a view toward practical applications, please add some explanations on the maximum number of simultaneous real-time nulling possible with the proposed algorithm that can be accommodated with a processing-time constraint in the SKA-low aperture array.

Author Response

Comments 1: Alpha, beta, and gamma in Equation (26) are presented without elaboration. Please add some explanation on Equation (26).

Response 1: Thank you for pointing this out. We agree with this comment.  We added more explanation on Equation (26), which can be found on line 231.

Comments 2. Authors presented a beamforming algorithm for an SKA-low station. Please explain how to extend the station-level algorithm to the 512 stations of the SKA1-low aperture array. Or rather explain how 512 stations work together.

Response 2: Thank you for pointing this out. We agree with this comment. Therefore, description about the multistage beamforming architecture is added on line 40 . As the tied-array beam is summed from multiple stations, the same algorithm can be used to the upper stage beamforming of 512 stations, where each station beam is treated as a single antenna element.

Comments 3. The foremost merit of the proposed algorithm is the processing speed. Please consider adding a scale of processing time in Figures 8 to 9.

Response 3: Thank you for pointing this out.  On line 421, descriptions about the processing speed of the FPGA are added to Chapter 4.5. For a FPGA using a specific clock frequency and number of parallelism, the processing speed of a single functional module is always proportional to the number of snapshots processed, so using the number of snapshots directly characterises the processing time.

Comments 4: Please check if the title of Figure 12 is correct.

Response 4: Thank you for pointing this out. We agree with this comment.  The title is modified, which could be found on line 434.

Comments 5: Figure 12:

1) An FPGA implementation of Figure 12 shows nulling for one angle while simulation results of Figures 5 to 7 show nulling at two angles. Please consider FPGA nulling at two angles.

2) The main beam change is appreciable in Figure 12 while it is negligible in Figures 5 to 7. Please add some comments.

Response 5: Thank you for pointing this out. We agree with this comment.  Therefore, we added Figure 13, which shows nulling at two angles. A natural paragraph has been added at the end of the chapter 4.5 with the relevant description, including the comments on the factors affecting the shape of main beam. Which can be found on line 442 to 452.

Comments 6: With a view toward practical applications, please add some explanations on the maximum number of simultaneous real-time nulling possible with the proposed algorithm that can be accommodated with a processing-time constraint in the SKA-low aperture array.

Response 6: Thank you for pointing this out. We agree with this comment. Therefore we added explanations about the maximum number of processing capability at the end of the chapter 4.5. The theoretical maximum number is M*(N-1), where M is the number of frequency bins and N is the number of antenna elements. It is recommended that M does not exceed 100 to save IO resources. Which could be found on line 452 to 456.

Round 2

Reviewer 2 Report

Comments and Suggestions for Authors

The authors have completed the required revisions, so I suggest accepting the article for publication.